# A Multimodal Desorption Electrospray Ionisation Workflow Enabling Visualisation of Lipids and Biologically Relevant Elements in a Single Tissue Section

**DOI:** 10.3390/metabo13020262

**Published:** 2023-02-11

**Authors:** Catia Costa, Janella De Jesus, Chelsea Nikula, Teresa Murta, Geoffrey W. Grime, Vladimir Palitsin, Véronique Dartois, Kaya Firat, Roger Webb, Josephine Bunch, Melanie J. Bailey

**Affiliations:** 1University of Surrey Ion Beam Centre, Guildford GU2 7XH, UK; 2Department of Chemistry, University of Surrey, Guildford GU2 7XH, UK; 3The National Physical Laboratory, Teddington TW11 0LW, UK; 4Center for Discovery and Innovation, Hackensack Meridian School of Medicine, Nutley, NJ 07110, USA

**Keywords:** multimodal imaging, correlative imaging, ion beam analysis, desorption electrospray ionisation mass spectrometry, biological tissue analysis

## Abstract

The colocation of elemental species with host biomolecules such as lipids and metabolites may shed new light on the dysregulation of metabolic pathways and how these affect disease pathogeneses. Alkali metals have been the subject of extensive research, are implicated in various neurodegenerative and infectious diseases and are known to disrupt lipid metabolism. Desorption electrospray ionisation (DESI) is a widely used approach for molecular imaging, but previous work has shown that DESI delocalises ions such as potassium (K) and chlorine (Cl), precluding the subsequent elemental analysis of the same section of tissue. The solvent typically used for the DESI electrospray is a combination of methanol and water. Here we show that a novel solvent system, (50:50 (%*v*/*v*) MeOH:EtOH) does not delocalise elemental species and thus enables elemental mapping to be performed on the same tissue section post-DESI. Benchmarking the MeOH:EtOH electrospray solvent against the widely used MeOH:H_2_O electrospray solvent revealed that the MeOH:EtOH solvent yielded increased signal-to-noise ratios for selected lipids. The developed multimodal imaging workflow was applied to a lung tissue section containing a tuberculosis granuloma, showcasing its applicability to elementally rich samples displaying defined structural information.

## 1. Introduction

Our understanding of biological processes is being continuously pushed by advances in multimodal imaging or spatial “omics”, which allows for the integration of structural, molecular and elemental information [1]. Several recent reports in the literature have explored the combination of mass spectrometry imaging (MSI) with elemental mapping techniques for biological applications [2,3,4,5,6,7]. The integration of these imaging modalities allows for spatial correlations of elements and their local molecular environments to be performed, further enhancing our knowledge of disease status and progression. A better understanding of pathogenesis at this level will enable the development of more efficient treatments [8,9,10,11,12]. 

Elemental analysis can reveal elemental dysregulation, accumulation or depletion, which are well described for many diseases such as Alzheimer’s disease [13], tuberculosis [14] and cancer [15,16,17]. Alkali metals in particular have been the subject of extensive research and are implicated in various neurodegenerative and infectious diseases as they are also known to disrupt lipid metabolism and play a key role in cell homeostasis [16,17,18,19]. Understanding the colocation of alkali metals with host biomolecules such as lipids and metabolites at the tissue level may shed new light on disease pathogenesis. Alkali metals and other trace elements can be imaged using elemental mapping techniques [7,20,21,22] such as laser ablation inductively couple mass spectrometry (LA-ICP-MS) [23,24,25,26,27,28,29], X-ray fluorescence (XRF) [30,31,32,33,34,35] and particle-induced X-ray emission (PIXE) [36,37,38,39,40,41], which have been extensively applied to the analysis of biological samples. 

There are a multitude of mass spectrometry imaging (MSI) techniques available [42,43,44] for imaging biological samples, including the commercially available desorption electrospray ionisation (DESI) [45,46,47,48,49], matrix-assisted laser desorption ionisation (MALDI) [50,51,52,53] and secondary ion mass spectrometry (SIMS) [54,55,56,57,58]. These techniques can provide images of the distribution of lipids, metabolites and (to a lesser extent) proteins [59,60]. Whilst these are primarily molecular analysis techniques, some reports in the literature have described how a limited amount of elemental information can be extracted using these techniques. SIMS can readily offer both molecular and elemental information, although the level of molecular information is highly dependent on the primary ion [58]. Additionally, the matrix effects preclude quantitative analysis [61,62]. MALDI is primarily used for molecular mapping, but Liu et al. reported on a workflow to target lipids, small metabolites and alkali ions (sodium and potassium) using this technique [63]. Most recently, a variant of DESI—nanoDESI—was used to simultaneously monitor metal ions (sodium and potassium) and metabolites [19]. Despite the seeming ability of these techniques to target both elements and molecules, the range of elements readily detectable and the quantitative capabilities are limited when compared to the primary elemental techniques listed above.

In most previous studies, sequential MSI and elemental mapping were performed on sequential tissue sections. A limitation of this approach is that smaller features are not accurately replicated in sequential sections, reducing the accuracy of feature colocation. It is therefore desirable to perform the two measurements sequentially using the same tissue sample. This is not trivial due to sample preparation incompatibility and modifications incurred to the samples by the preceding analysis. For example, it is reasonable to expect that elemental mapping using any of the techniques described above would modify the chemistry of the sample due to localised heating, and indeed this has been previously demonstrated in the case of PIXE [64,65]. It is therefore desirable to develop molecular imaging strategies that do not delocalise elemental species. In previous work, it was shown that DESI imaging, employing a MeOH:H_2_O electrospray solvent, preserved the location and concentration of various elemental species in lung tissues, for example Fe and S. However, elements such as Cl and K were delocalised by the DESI analysis and could not be imaged.

DESI offers the prospect of analysis under ambient conditions, meaning that the sample is analysed in its native state. It offers further advantages in terms of analyte coverage for lipids and metabolites [66]. Here we report on a novel DESI electrospray solvent system (MeOH: EtOH), which enables subsequent elemental mapping using the same tissue section without the delocalisation of Cl and K and other detected trace elements. This solvent combination was previously used by Lewis et al., who used direct analyte probed nanoextraction (DAPNe) for the extraction and analysis of lipids from tissue sections with no detrimental effect on the subsequent elemental imaging, but to our knowledge, this has not been used before with DESI [67]. 

The ion beam analysis (IBA) techniques PIXE and Elastic Backscattering Spectrometry (EBS) were used to monitor the delocalisation and loss of elements following DESI imaging. To assess the performance of the MeOH:EtOH solvent system to image the molecular species, a benchmarking experiment was performed to compare the new MeOH:EtOH to the widely used MeOH:H_2_O solvent system [64,65]. This was carried out by comparing the intensities and coverage of the lipid peaks observed in homogenised liver. The analytical workflow was then applied to a lung tissue section containing a tuberculosis lesion, enabling an integrated analysis of alkali ions, transition metals, halogens and lipids.

## 2. Materials and Methods

### 2.1. Sample Preparation

#### 2.1.1. Homogenized Tissue

Rat liver homogenates were prepared as described by Swales et al. [68] The liver tissue was homogenised and pipetted into moulds (2 mL bottom end of Pasteur pipette bulb). The homogenates were snap frozen in propanol and then iso-pentane and were stored in −80 °C. Three homogenates were sectioned sequentially at 10 µm thickness using a Thermo NX70 Cryostar (Thermo Fisher Scientific, Bremen, Germany) and were thaw mounted onto a 1.4 μm thick polyethylene (PET) substrate (Leica, Wetzlar, Germany). The slides were analysed sequentially using DESI and PIXE/EBS in the areas highlighted in Appendix A. All animals and tissue were managed in accordance with the UK Home Office Animals (Scientific Procedures) Act 1986. The organs used within this study were within the 3Rs principles as they comprised a control material surplus to the original study for which they were intended. 

#### 2.1.2. Snap-Frozen Lung Tissue

Rabbit infection and sample collection were performed in Biosafety Level 3 (BSL3) facilities and approved by the Institutional Animal Care and Use Committee of the National Institute of Allergy and Infection Disease, NIH, Bethesda, MD, USA (Protocol number LCIM-3). All studies followed the guidelines and basic principles stated in the United States Public Health Service Policy on Humane Care and Use of Laboratory Animals. All samples collected from *Mycobacterium tuberculosis*-infected animals were handled and processed in the BSL3 in compliance with protocols approved by the Institutional Biosafety Committee of the National Institute of Allergy and Infection Disease, NIH, and Hackensack Meridian Health, NJ, USA.

Female New Zealand White (NZW) rabbits weighing 2.2–2.6 kg were maintained under specific pathogen-free conditions and fed water and chow ad libitum. The NZW rabbit ID 713 was infected with *M. tuberculosis* HN878 using a nose-only aerosol exposure system as described [69]. At 14 weeks postinfection, once mature cellular and necrotic lung lesions had developed, lung lesions embedded in the surrounding tissue were collected for imaging and were snap frozen (unprocessed) in liquid nitrogen vapor as described previously [70,71]. To sterilize the samples and inactivate all viable *M. tuberculosis* bacilli, samples were γ-irradiated in a Co-60 irradiator until exposure reached 3 Mrad (validated as a sufficient exposure to kill all viable *M. tuberculosis* bacteria present in lung lesions). Dry ice was resupplied as required to keep the samples always frozen. The frozen rabbit lesions were sectioned at a 10 μm thickness using a CM1860 UV cryostat (Leica, Wetzlar, Germany) at −20 °C. The sections were thaw mounted onto 1.4 μm thick poly(ethylene terephthalate) (PET) membrane slides (Leica, Wetzlar, Germany), shipped on dry ice and stored at −80 °C.

### 2.2. DESI Imaging 

DESI was used to image small molecules in the tissue homogenates prior to the ion beam analysis. A prototype DESI source with a recessed capillary (Waters, Wilmslow, UK) was coupled to a Xevo G2-XS (Waters, Wilmslow, UK) mass spectrometer. A 95:5 (%*v*/*v*) methanol (MeOH)/water (H_2_O) or 50:50 (%*v*/*v*) MeOH/ethanol (EtOH) spray solvent was delivered at a rate of 2 µL/min using an Ultimate 3000 UHPLC system (Thermo Fisher Scientific, Bremen, Germany) with an electrospray voltage of 0.6 kV and ion block temperature set to 100 °C. Prior to acquisition, mass calibration in positive ion mode was performed using a polylactic acid (PLA) sublimed slide made in house with a collision energy of 35 V. Data were acquired in positive ion “sensitivity” mode, with a mass range of *m*/*z* 100–1200 at a calculated mass resolving power of 15,000 at *m*/*z* 200. The tissue region for imaging was selected using High-Definition Imaging (Waters, Wilmslow, UK) software. The nominal pixel size was 75 × 75 µm using a stage speed of 150 µm/s, acquiring the data at 2 pixels/s. 

#### Data Analysis—DESI

Waters RAW data files were converted into imzML files through a two-step conversion. The first was the conversion to mzML using Proteowizard [72], then to an imzML using the imzML converter [73]. The imzML data were analysed using Spectral Analysis [74] (version 1.4.0) and run using MATLAB (version 2018b). Prior to generating a mean spectrum, the data were preprocessed using a rebinning method (bin size of 0.001) to generate the mean spectra, and then they were normalised to the total ion intensity when generating the datacube. 

A lipid peak list (top 50 most intense peaks assigned as lipids) was generated through the tentative assignment of *m*/*z* peaks detected in the liver homogenates (see Appendix A) using in-house MATLAB scripts which matched the data against the Human Metabolome Database (HMDB) [75]. The peak assignment was achieved using a +/−15 ppm mass match and through the inspection of the DESI ion images to ensure that the signals originated from the sample and not the background. Peak assignments were further checked against criteria such as abundance in mammalian tissue, likelihood of adduct formation and likelihood of ion formation in positive ion mode.

### 2.3. Ion Beam Analysis

After DESI, the samples were simultaneously analysed by proton induced X-ray emission (PIXE) and elastic backscattered spectrometry (EBS) using a 2 MV Tandem accelerator (High Voltage Engineering, Amersfoort, The Netherlands). The samples were placed in a vacuum chamber pumped to 10^−6^ mBar and irradiated using 2.5 MeV protons with beam currents ranging from 300–600 pA. The beam was focused to approximately 2 × 2 µm (measured using a 75 × 75 µm 1000 copper grid). The scan size was 1 × 1 mm with a pixel dwell time set at 0.3 ms. X-rays were detected using a silicon drift detector (SDD) fitted with a 130 µm Be filter, mounted at an angle of 135° to the beam direction in the horizontal plane. Backscattered particles were simultaneously collected and detected using a PIPS detector with an active area of 150 mm^2^, placed 52.5 mm away from the sample and mounted at a 25° exit angle. 

The liver homogenate samples were analysed until a charge of 2000 nC was collected in the areas highlighted in Appendix A. Three sequential rabbit lesion sections were analysed in mosaic scan mode (2 × 2 squares of 1 × 1 mm each), with each square being analysed until 4000 nC of charge were collected. 

#### Data Analysis—Ion Beam Analysis

The X-ray and backscattered particle spectra were calibrated using a BCR-126A lead glass standard. The data were acquired and analysed using OMDAQ-3 software (Oxford Microbeams, Ltd., Oxfordshire, UK) [76].

## 3. Results 

### 3.1. Homogenized Tissue

As highlighted in Appendix A, selected regions of three sequential liver homogenate sections were analysed by DESI using two different solvent systems: 95:5 (%*v*/*v*) MeOH/H_2_O or 50:50 (%*v*/*v*) MeOH/EtOH. Each section was then imaged by PIXE/EBS with an ROI encompassing the interface of the areas sampled by DESI to observe delocalisation. 

A sequential PIXE analysis at the interface of the areas previously analysed by DESI clearly demonstrated that the MeOH:H_2_O solvent caused a visible loss of mobile ions such as chlorine and potassium as well as phosphorus, as shown in Figure 1A. Conversely, the MeOH:EtOH solvent caused no measurable loss or delocalisation of any elements measured by PIXE (Figure 1B). This was further supported by Appendix A, showing the overlay of the PIXE spectra taken from regions of interest derived from areas with/without prior DESI analysis, and by Figure 1C, which shows the signal loss for several elements after DESI analysis. Statistical tests (*t*-tests) showed that the differences observed in the PIXE peak area ratios measured between the two solvents were statistically significant for all elements monitored (see Appendix A).

Appendix A shows example DESI spectra taken with each solvent system from the background (PET substrate) and shows that the MeOH/EtOH mixture generated less intense peaks from the PET background. Figure 2 compares the mean spectra collected from tissue ROIs corresponding to analysis via the two solvent mixtures. The spectra were broadly similar, with the 50:50 (%*v*/*v*) MeOH:EtOH solvent producing slightly higher TIC-normalised peak intensities than MeOH:H_2_O, especially in the higher *m*/*z* ranges, possibly due to the higher solubility of lipids in the purely organic solvent. 

The top 50 most intense lipid peaks were selected, and a list of measured *m*/*z* and peak assignments are listed in Appendix A. The breakdown per assigned lipid class (Figure 3A) and per *m*/*z* range (Appendix A) were compared for each spray solvent. The intensities of the 10 most intense peaks for each solvent were compared in Figure 3B,C. As expected, these were comprised predominantly of peaks assigned to phosphocholines (PC), due to their prevalence in cell membranes. The two datasets shared 9 out of the 10 lipid peaks (marked by the asterisks) and these were detected with statistically significant higher peak intensities (*p* < 0.05; see Appendix A) when the MeOH:EtOH solvent was used. Ion images were generated for each of the 10 peaks, and all were shown to originate from the tissue homogenate rather than the background (Appendix A). Appendix A shows the peak intensity measured for the remaining 40 top lipid peaks for each solvent. These observations, as well as Figure 3, demonstrate that the two solvents did not preferentially target lipids in one particular *m*/*z* region of the mass spectrum.

A brief study of the adducts formed using the two solvent systems showed that MeOH:EtOH produced more [M+K]^+^ adducts in the top 50 lipids than MeOH:H_2_O (see Appendix A). This is interesting as the PIXE data showed that MeOH:EtOH did not cause a loss or delocalization of potassium, offering a possible explanation for the increased proportion of intense lipid peaks appearing as [M+K]^+^. In contrast, the ratio [M+H]^+^/[M+K]^+^ for a selection of PC lipids was found not to change significantly between the two solvent systems (see Appendix A). Future work could explore whether the addition (and preservation) of K in biological samples can be used to enhance lipid coverage.

The data shown above demonstrate that using 50:50 (%*v*/*v*) MeOH:EtOH as a DESI electrospray solvent can be advantageous for increasing the sensitivity to lipids. Additionally, the data showed that the new solvent did not cause the removal or delocalisation of mobile ions as measured by PIXE. To confirm this observation, and to demonstrate the ability of MeOH:EtOH to provide images, these two solvent systems were used to analyse the snap-frozen lung tissue sections. 

### 3.2. Snap-Frozen Lung Tissue from Rabbits

The optical images taken before the DESI analysis of the of snap-frozen rabbit lung tissues containing a caseous granuloma (a lesion caused by tuberculosis) is shown in Appendix A. Two of the sections were first analysed using the two DESI solvents and a third were left untouched to be used as a control for ion beam elemental mapping. The regions were chosen to include three regions of the granuloma—the caseum or necrotic centre, cellular rim and uninvolved lung. 

Figure 4 shows the red, green and blue (RGB) overlay of *m*/*z* 953 (TG (58:8) [M+Na]^+^), *m*/*z* 832 (PC (38:4) [M+K]^+^) and *m*/*z* 780 (PC(36:5) [M+H]^+^) obtained using DESI and two spray solvents—95:5 (%*v*/*v*) MeOH:H_2_O and 50:50 (%*v*/*v*) MeOH:EtOH. These peaks were chosen because they very clearly show the different pathology regions of the granuloma tissue. In both overlays, the different regions of the granulomas were visible, as presented in Appendix A. Figure 4 demonstrates that the MeOH:EtOH spray solvent could produce images of comparable quality to the more widely used MeOH:H_2_O.

The chlorine (Cl) and potassium (K) elemental maps from a sequential PIXE analysis of the three tissue sections are shown in Figure 5. As shown in the maps in Figure 5, there was a loss of Cl and K when MeOH:H_2_O was used as a spray solvent (*p* > 0.05; see Appendix A), but Fe remained unchanged. Appendix A shows the remaining elemental maps for phosphorus (P), sulphur (S) and the total EBS map.

As each of the lung sections were analysed in mosaic scan mode (four squares arranged in a 2 × 2 array, each square of dimensions 1 × 1 mm), there were four analysis regions per sample. Appendix A shows the overlay of the X-ray spectra for one of the squares—the one in each sample which captured the caseum. There is a clear loss of K and Cl when MeOH:H_2_O was used, in agreement with the elemental maps shown in Figure 5. As observed with the tissue homogenate sections, there was a small loss of P when the MeOH:H_2_O solvent was used, confirmed by *t*-test (*p* < 0.05, see Appendix A). On the other hand, the difference in P peak area between the control sample and the sample analysed using MeOH:EtOH was not statistically different (*p* > 0.05). This confirmed that MeOH:EtOH could preserve the elemental integrity of the sample. 

## 4. Discussion

The data presented here demonstrates that elemental mapping can be carried out following DESI imaging using a 50:50 (%*v*/*v*) MeOH:EtOH spray solvent. To our knowledge, this is the first time that this solvent system has been reported for use in a DESI system, although it is noteworthy that MeOH:EtOH has been used previously by Lewis et al. for liquid extraction surface analysis to enable the extraction of lipids from tissue sections [67]. Our data showed that this novel solvent system can produce images of similar quality (Figure 4) to the more conventional MeOH:H_2_O solvent system, with similar coverage across the mass spectrum (Appendix A) and enhanced an signal-to-noise ratio for selected peaks (Figure 3). The increased sensitivity towards lipids using this alcohol-based spray solvent may be related to the known effect of alcohols on lipid bilayer properties. Alcohols, and in particular ethanol and methanol, are known to change the structural properties of membranes, increasing their permeability. In this instance, it is expected that the employment of these solvents for DESI analysis would increase the sensitivity towards lipids [77]. Furthermore, no delocalisation or loss of elements was observed using PIXE/EBS. This is presumably because mobile ions such as potassium and chlorine are highly water soluble and so delocalise in the conventional MeOH:H_2_O solvent system, but not in the aqueous-free novel solvent mixture. 

We have therefore demonstrated the successful multimodal imaging of molecular and elemental markers on a single tissue section, overcoming issues such as changes incurred to the samples by preceding measurements. In this work we have shown that IBA can be carried out following DESI analysis, but this observation is also relevant to other elemental imaging techniques such as XRF, SEM and LA-ICP-MS, for example. 

A limitation of the current approach is that the image resolution of commercial DESI instrumentation (typically 50–100 microns) is inferior to many elemental imaging modalities, which can be submicron, depending on the technique. Elemental mapping techniques can resolve single cells, whereas most DESI systems currently cannot. Being able to resolve single cells would improve the certainty of correlation between molecular and elemental images. Recent developments in DESI source technology should drive substantial improvements to image resolution and enable imaging at comparable pixel sizes in the near future [78].

Future work should investigate different ratios of methanol to ethanol and whether this changes the sensitivity/coverage of the DESI analysis and/or the elemental composition of a sample. Additionally, it would be useful to use PIXE to establish whether the absence of potassium caused by the MeOH:H_2_O solvent prevents [M+K]^+^ adduct formation, thereby reducing sensitivity to certain lipids. The preservation (or even addition) of K in biological samples may be useful to enhance lipid coverage.

## 5. Conclusions

This work has demonstrated that the novel DESI spray solvent 50:50 (%*v*/*v*) methanol:ethanol enables sequential elemental mapping out on the same tissue section. This is desirable to allow the accurate correlation of elemental and molecular features since regions of interest are not always accurately reproduced in sequential sections. In this work, the new solvent system produced similar, if not better, lipid coverage and sensitivity in positive ion mode when compared to the conventional methanol:water solvent. This work therefore demonstrates the successful multimodal imaging of molecular and elemental markers on a single tissue section, overcoming issues such as changes incurred to the samples by preceding measurements.

## Figures and Tables

**Figure 1 metabolites-13-00262-f001:**
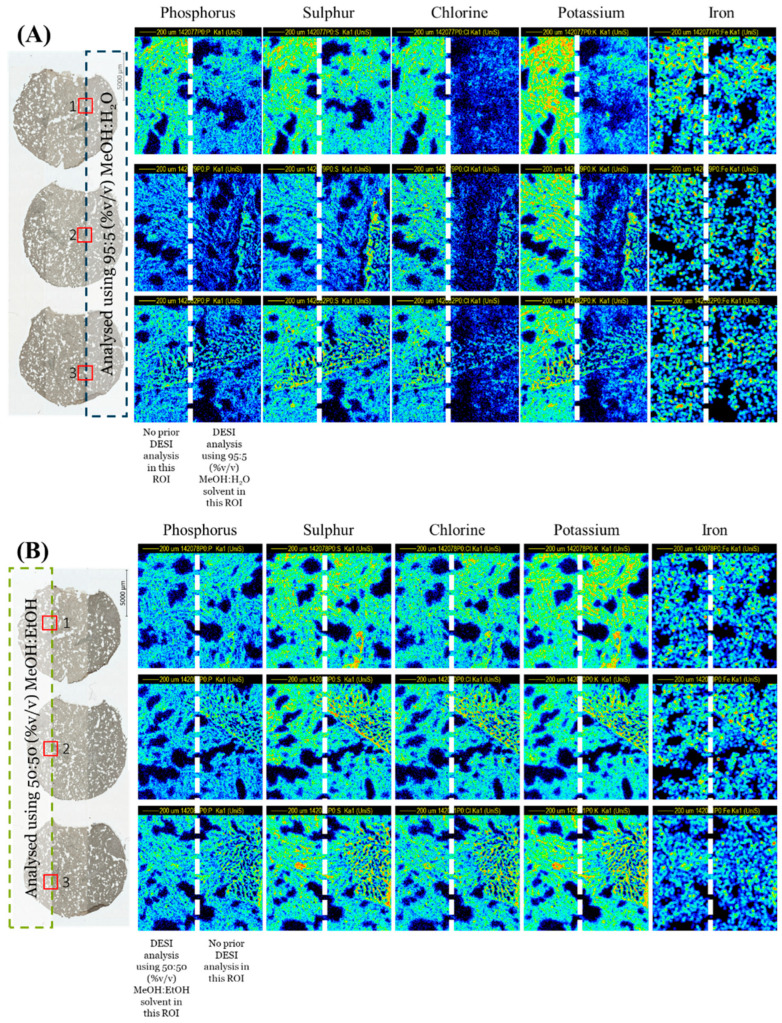
PIXE maps taken post-DESI from 3 sequential liver homogenate sections using an ROI scanned over the edge of the DESI ROI using (**A**) 95:5 (%*v*/*v*) MeOH:H_2_O and (**B**) 50:50 (%*v*/*v*) MeOH:EtOH. The DESI (dotted line) and PIXE ROIs (solid line) are indicated on the far-left image. (**C**) Average ratio (*n* = 3) of the elemental peak areas measured by PIXE from ROIs with/without prior DESI. See Appendix A for further information on the analysis locations for each technique.

**Figure 2 metabolites-13-00262-f002:**
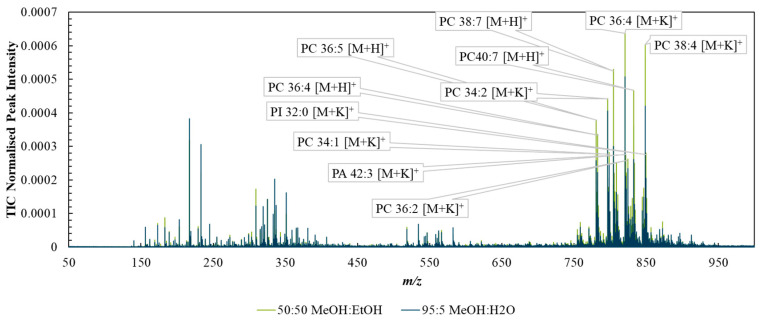
Spectra taken from regions of interest capturing tissue-only areas acquired with DESI using the two solvent systems—95:5 (%*v*/*v*) methanol/water (blue) or 50:50 (%*v*/*v*) methanol/ethanol (green).

**Figure 3 metabolites-13-00262-f003:**
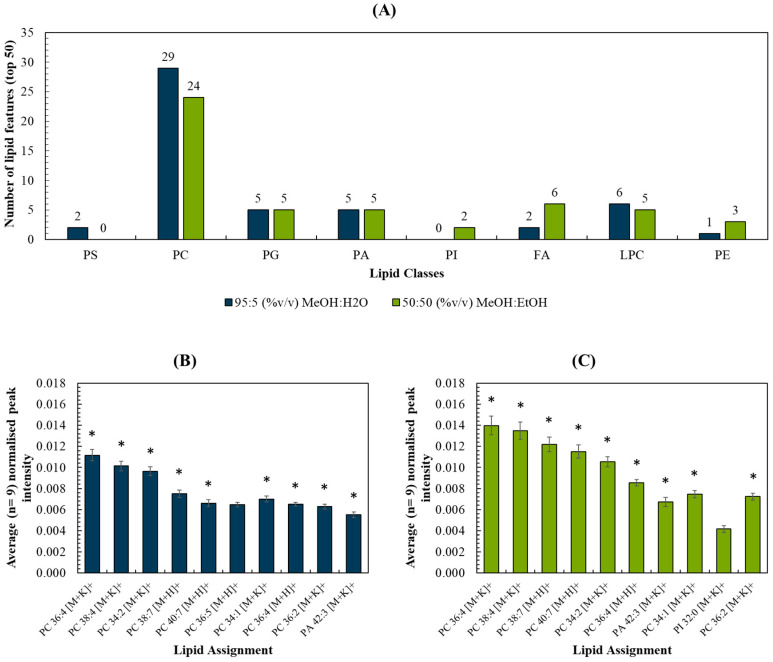
(**A**) Number of lipid features (out of the top 50) detected per lipid classes for MeOH:H_2_O and MeOH:EtOH spray solvents, respectively; (**B**,**C**) Top 10 most abundant peaks and their respective intensities (normalised to the total ion count) for (**B**) MeOH:H_2_O and (**C**) MeOH:EtOH. * refers to peaks detected using both solvents in liver homogenates.

**Figure 4 metabolites-13-00262-f004:**
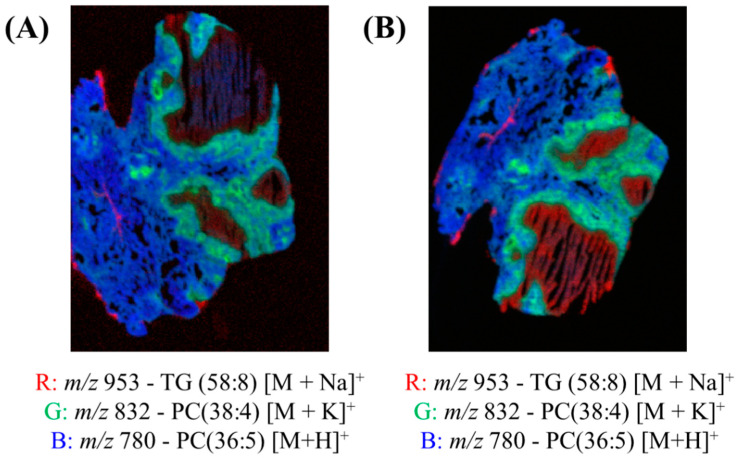
Red, green and blue (RGB) overlay of *m*/*z* 953 (TG (58:8) [M+Na]^+^), *m*/*z* 832 (PC (38:4) [M+K]^+^) and *m*/*z* 780 (PC(36:5) [M+H]^+^) obtained using DESI and two spray solvents—(**A**) 95:5 (%*v*/*v*) MeOH:H_2_O and (**B**) 50:50 (%*v*/*v*) MeOH:EtOH.

**Figure 5 metabolites-13-00262-f005:**
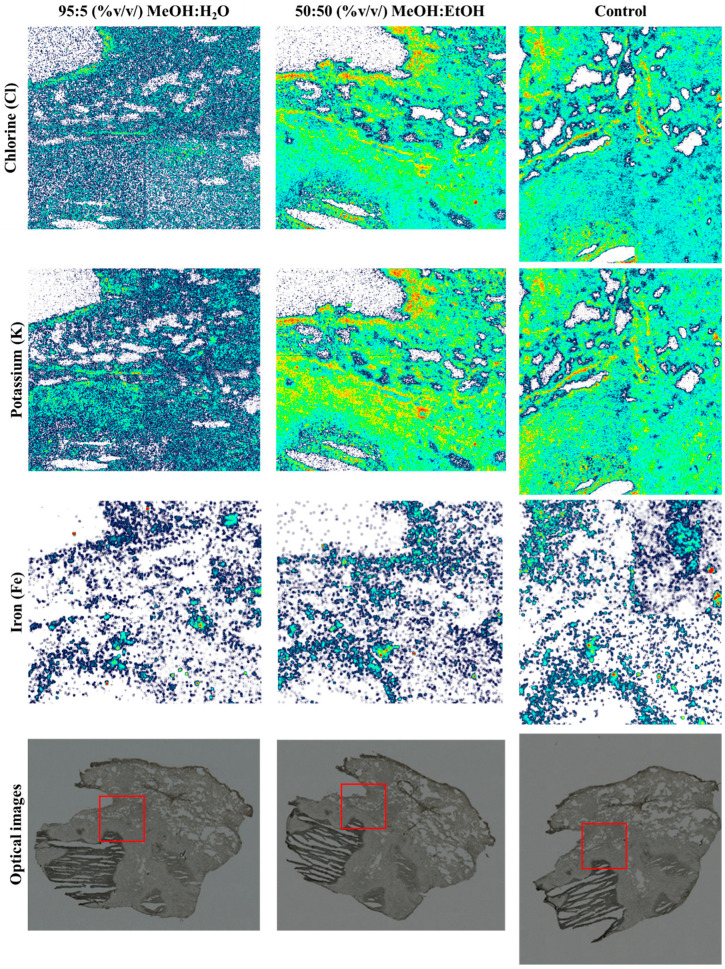
Chlorine (Cl), potassium (K) and iron (Fe) PIXE maps taken from snap-frozen lung tissue sections after DESI analysis using 95:5 (%*v*/*v*) MeOH:H_2_O or 50:50 (%*v*/*v*) MeOH:EtOH. A third section (control) was also analysed—no DESI measurements were taken on this sample. Optical images taken before analysis highlighting the PIXE/EBS analysis areas (red squares) on each tissue.

## Data Availability

All datasets generated for this study are included in the article.

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
