# Peer review of "A Multimodal Desorption Electrospray Ionisation Workflow Enabling Visualisation of Lipids and Biologically Relevant Elements in a Single Tissue Section"

_metabolites, 2023, doi:10.3390/metabo13020262_

Round 1

Reviewer 1 Report

The submitted paper proposes MeOH/EtOH as a new electrospray solvent suitable for desorption electrospray ionization (DESI) imaging and subsequent particle-induced X-ray emission (PIXE) analysis and states that MeOH/EtOH has the advantage of improved detection sensitivity in DESI imaging and avoiding delocalization of ions such as potassium and chlorine compared to the conventional electrospray solvent MeOH/H2O. Therefore, the submitted paper would be suitable for publication in Metabolites. However, there are some points to be considered before the publication. Please consider and revise the manuscript according to the comments listed below.

(1)   How MeOH/EtOH was found to be the suitable electrospray solvent was not fully explained. Please add an explanation of on what basis, or based on previous studies, the authors thought MeOH/EtOH could be used as an electrospray solvent, which enables subsequent elemental mapping without delocalization of ions.

(2)   Related to the above question, why is MeOH/EtOH able to preserve the elemental integrity of the sample, unlike MeOH/H2O? Please add a discussion of this reason.

(3)   The mixture of 50:50 (%v/v) MeOH/EtOH was used as the electrospray solvent in the submitted study. Did the authors investigate the effect of the ratio of MeOH to EtOH in the solvent on the peak intensities in the DESI and delocalization of ions?

(4)   Does the intensity of the molecular ion peak of [M+K]+ fluctuate with the application of the MeOH/EtOH that does not cause removal or delocalization of potassium ions?

(5)   It is difficult to distinguish between the two spectra in Figure S3, so please change the color and contrast.

(6)   The manuscript contains a number of "Reference source not found" errors, which make the content unclear. In particular, it was unclear what the (C) in line 249 was pointing out. Please correct them correctly.

(7)   In lines 211-212, “the 50:50 (%v/v) MeOH:EtOH solvent producing slightly higher TIC-normalised peak intensities” would be “the 50:50 (%v/v) MeOH:EtOH solvent producing slightly higher TIC-normalised peak intensities than MeOH:H2O”.

Author Response

We would like to thank the reviewer for taking the time to read our manuscript and provide us with some useful suggestions. Below we provide a point-by-point answer to the reviewer’s comment.

The submitted paper proposes MeOH/EtOH as a new electrospray solvent suitable for desorption electrospray ionization (DESI) imaging and subsequent particle-induced X-ray emission (PIXE) analysis and states that MeOH/EtOH has the advantage of improved detection sensitivity in DESI imaging and avoiding delocalization of ions such as potassium and chlorine compared to the conventional electrospray solvent MeOH/H2O. Therefore, the submitted paper would be suitable for publication in Metabolites. However, there are some points to be considered before the publication. Please consider and revise the manuscript according to the comments listed below.

  • How MeOH/EtOH was found to be the suitable electrospray solvent was not fully explained. Please add an explanation of on what basis, or based on previous studies, the authors thought MeOH/EtOH could be used as an electrospray solvent, which enables subsequent elemental mapping without delocalization of ions.

In the discussion, we briefly mention a paper by Lewis et al. [Ref #67; lines 288-289], which observed that this solvent mixture did not cause loss or delocalisation of mobile ions in similar tissue sections. This paper used a different technique (DAPNe) so it was unclear whether the same effect would be applicable to DESI, hence our investigations presented in this paper. We have added a sentence to the Introduction that explicitly says this.

  • Related to the above question, why is MeOH/EtOH able to preserve the elemental integrity of the sample, unlike MeOH/H2O? Please add a discussion of this reason.

This is a very interesting question. We hypothesise that this is due to the lack of water in the spray solvent. Ion such as K and Cl are highly water soluble and without water in the spray, they are likely not easily desorbed and ionised using DESI. We have added a line to the discussion (297-298) explaining this.

  • The mixture of 50:50 (%v/v) MeOH/EtOH was used as the electrospray solvent in the submitted study. Did the authors investigate the effect of the ratio of MeOH to EtOH in the solvent on the peak intensities in the DESI and delocalization of ions?

We did not investigate any other ratios of MeOH:EtOH. This would make for good future work in this area and we have alluded to this in the Discussion.  

  • Does the intensity of the molecular ion peak of [M+K]+fluctuate with the application of the MeOH/EtOH that does not cause removal or delocalization of potassium ions?

This is an excellent point. We have briefly looked at the adduct formation with K and added an extra Figure (Figure S7) to the Supporting Information showing the distribution of adducts formed in the top 50 detected lipids. It is interesting that MeOH:EtOH presents a higher number of [M+K]+ adducts formed than MeOH:H2O and we have remarked this in the text (see paragraph before Fig.3 ). More work is necessary to further explore this (also added to the Discussion).

  • It is difficult to distinguish between the two spectra in Figure S3, so please change the color and contrast.

Thank you. Figure S3 has been modified to make it clearer.

  • The manuscript contains a number of "Reference source not found" errors, which make the content unclear. In particular, it was unclear what the (C) in line 249 was pointing out. Please correct them correctly.

Line 249 should refer to Figure 5. This has been corrected. 

  • In lines 211-212, “the 50:50 (%v/v) MeOH:EtOH solvent producing slightly higher TIC-normalised peak intensities” would be “the 50:50 (%v/v) MeOH:EtOH solvent producing slightly higher TIC-normalised peak intensities than MeOH:H2O”.

Thank you. This has been corrected.

Reviewer 2 Report

Overall, this is a well-written and interesting manuscript. However there are several points that the authors need to address in a revised manuscript. These points are listed below in order of appearance in the paper.

1. Page 2, lines 78-82: Previous work is mentioned, but no reference is cited. The work should be cited in the references.

2. Page 4, lines 155-162: Here the authors are going over the Data Analysis. The liver homogenates are matched to the Human Metabolome Database. However, the homogenates are from rat liver. Some rationale needs to be provided why this comparison is valid.

3. Lines 191, 195, 209, 219, 222, 229, 249, 252, 265, 266, 273: The lines have an "Error! Reference source not found." statement. Please check the citation manager you are using.

4. Pages 5 and 6: Figure 1: In Figure 1a and 1b, the authors use "No DESI analysis" labels beneath the images. It might be better to replace the label with something like "PIXE analysis" to improve the readability of the figure.

Once these issue have been addressed, this well done study should be in a suitable form for publication.

Author Response

We would like to thank the reviewer for reading the manuscript. Below you will find a point-by-point answer to the comments.

Overall, this is a well-written and interesting manuscript. However there are several points that the authors need to address in a revised manuscript. These points are listed below in order of appearance in the paper.

  1. Page 2, lines 78-82: Previous work is mentioned, but no reference is cited. The work should be cited in the references.

Thank you. References have now been added.

  1. Page 4, lines 155-162: Here the authors are going over the Data Analysis. The liver homogenates are matched to the Human Metabolome Database. However, the homogenates are from rat liver. Some rationale needs to be provided why this comparison is valid.

Databases for animal tissue are rare and therefore HMDB is commonly used for tentative peak annotation in animal models, especially mammalian tissues. A few examples where this has been done: https://doi.org/10.1007/s11306-012-0430-8, https://doi.org/10.1021/ac4040967, https://doi.org/10.1186/s12944-017-0633-0.

  1. Lines 191, 195, 209, 219, 222, 229, 249, 252, 265, 266, 273: The lines have an "Error! Reference source not found." statement. Please check the citation manager you are using.

Thank you. This occurred when the manuscript was moved to the MDPI format. All of these have been corrected now. Please note these were not bibliographic references but rather cross-references to the Figures in the paper.

  1. Pages 5 and 6: Figure 1: In Figure 1a and 1b, the authors use "No DESI analysis" labels beneath the images. It might be better to replace the label with something like "PIXE analysis" to improve the readability of the figure.

This Figure has been updated to hopefully make it clear.

Once these issue have been addressed, this well done study should be in a suitable form for publication.

Reviewer 3 Report

Dear Authors, I found the paper to be overall very well written and I felt confident that you performed careful research data interpretation. However, I recommend that a minor revision of the manuscript is warranted and I explain my concerns below. Please use same verbal mode - normally the past tense is used because the experimental and interpretation part has already taken place. e.g. in successive sentences like lines 218-220 you have used both past and present! Line 221- “The intensities of the 10 most intense peaks for each solvent are compared” – I would like to see the selected lipid masses in the mass spectra, Fig. 2. Even if the results are conclusive, the introduction and conclusions should be correlated with what is already known in the literature about the effect of alcohols on Lipid Bilayer Properties Please correct the “Error! Reference source not found” for indicated the Figures in the text. Please correct H2O – in several line (236, 281, etc) 2 is not subscript !

Author Response

Dear Authors, I found the paper to be overall very well written and I felt confident that you performed careful research data interpretation. However, I recommend that a minor revision of the manuscript is warranted and I explain my concerns below.

We would like to thank the reviewer for taking the time to thoroughly read our manuscript and giving us the comments below.

Please use same verbal mode - normally the past tense is used because the experimental and interpretation part has already taken place. e.g. in successive sentences like lines 218-220 you have used both past and present!

Thank you. We have gone through the manuscript and corrected this where appropriate.

Line 221- “The intensities of the 10 most intense peaks for each solvent are compared” – I would like to see the selected lipid masses in the mass spectra, Fig. 2.

Thank you. I have now labelled the peaks in Fig. 2 that correspond to the top 10 lipids detected.

Even if the results are conclusive, the introduction and conclusions should be correlated with what is already known in the literature about the effect of alcohols on Lipid Bilayer Properties.

Thank you for bringing this up to our attention. We have now added a sentence to the Discussion relating our findings to the known effects of alcohols on the lipid bilayers.

Please correct the “Error! Reference source not found” for indicated the Figures in the text. Please correct H2O – in several line (236, 281, etc) 2 is not subscript !

Thank you, we have now corrected this.

Round 2

Reviewer 1 Report

The author revised the manuscript appropriately according to my comments, and the question was resolved. Therefore, the revised manuscript will be accepted for publication in its present form.

Reviewer 2 Report

The authors have adequately addressed the concerns of the reviewer. They have also improved the readability of the manuscript.